# E3 Ubiquitin Ligase *TRIP12* Controls Exit from Mitosis via Positive Regulation of *MCL-1* in Response to Taxol

**DOI:** 10.3390/cancers15020505

**Published:** 2023-01-13

**Authors:** Kripa S. Keyan, Rania Alanany, Amira Kohil, Omar M. Khan

**Affiliations:** Division of Biological and Biomedical Sciences, College of Health and Life Sciences, Hamad Bin Khalifa University, Doha P.O. Box 34110, Qatar

**Keywords:** *FBW7*, *MCL-1*, chemotherapy, Taxol, proteasomal degradation, mitotic arrest

## Abstract

**Simple Summary:**

Taxol is a chemotherapy drug used in treatment of multiple cancers. Taxol works by blocking an essential process of cell division called mitosis. Although, Taxol treatment shows promise in fight against cancer, many patients eventually develop resistance. Loss of function mutations in an E3 ubiquitin ligase component *FBW7*, are casually associated with cancer chemotherapy- and Taxol resistance. *FBW7* is essential for degradation of a pro-survival protein *MCL-1*. In the absence of *FBW7*, *MCL-1* protein accumulates, and cancer cells escape Taxol induced death. In this work we discover that another E3 ubiquitin ligase *TRIP12* is required by cancer cells for efficient mitosis and completion of cell division. Inhibition of *TRIP12* enhances Taxol induced cell death in an *FBW7* and *MCL-1* dependent manner. Thus, *TRIP12*/*FBW7*/*MCL-1* axis is an important determinant of Taxol response in cancer cells.

**Abstract:**

Chemotherapy resistance is a major hurdle in cancer treatment. Taxol-based chemotherapy is widely used in the treatment of cancers including breast, ovarian, and pancreatic cancer. Loss of function of the tumor suppressor F-box WD-40 domain containing 7 (*FBW7*) mutations leads to the accumulation of its substrate *MCL-1* which is associated with Taxol resistance in human cancers. We recently showed that E3 ubiquitin ligase *TRIP12* is a negative regulator of *FBW7* protein. In this study, we find that Taxol-induced mitotic block in cancer cells is partly controlled by *TRIP12* via its positive regulation of *MCL-1* protein. Genetic inhibition of *TRIP12* accelerates *MCL-1* protein degradation in mitosis. Notably, introducing double-point mutations in lysines 404/412 of *FBW7* to arginine which makes it resistant to proteasomal degradation, leads to the sharp reduction of *MCL-1* protein levels and sensitizes cancer cells to Taxol-induced cell death. Finally, *TRIP12* deletion leads to enhanced mitotic arrest and cell death in an *FBW7* and *MCL-1* dependent manner in multiple cell lines including colorectal and ovarian cancer but not in breast cancer. Thus, the *TRIP12*/*FBW7*/*MCL-1* axis may provide a therapeutic target to overcome Taxol-associated chemotherapy resistance in cancer.

## 1. Introduction

Cancer is a major human health challenge and the leading cause of death worldwide [1]. Late detection, widespread metastasis, and de novo and acquired chemotherapy resistance are the major challenges in cancer management. Taxol or texane-based chemotherapy drugs are widely used in cancer management including breast, ovarian, and pancreatic cancer, either alone or in combination with adjuvant therapy [2]. Taxol is a mitotic spindle stabilizer which arrests cells in mitosis thereby inducing cell death [3,4]. Although Taxol has been beneficial in cancer treatment, many patients’ cancer eventually relapses with near complete resistance.

*FBW7* is a substrate adaptor of the Cullin-ring SKP1/CUL1/F-Box (SCF)-type E3 ubiquitin ligase complex. *FBW7* targets oncogenic proteins including c-MYC, CyclinE, c-JUN, Notch1, and *MCL-1* for proteasomal degradation [5]. Consequently, loss of *FBW7* function mutations leads to accumulation of some of those proteins which is associated with human cancers and chemotherapy resistance [6,7]. For example, *FBW7* mediated proteasomal degradation of *MCL-1* in mitosis is a major cell death mechanism in response to Taxol and apoptosis inducing agents [6,7]. Thus, *FBW7* mutations confer Taxol resistance to cancer cells due to accumulation of its anti-apoptotic substrate *MCL-1* [6,7]. In addition to somatic mutations, *FBW7* protein is downregulated post translationally in cancer [8,9]. Thus, understanding pathways and mechanisms that converge on *FBW7* protein activity and stability can help unravel novel targets to tackle cancer-associated chemotherapy resistance.

The thyroid hormone receptor interactor protein 12 (*TRIP12*), also known as the E3 ubiquitin ligase for Arf (ULF), is a HECT-domain E3-ligase. The yeast homologue of *TRIP12* protein, Ufd4, was identified as an ubiquitin ligase which could extend polyubiquitin chains on a protein substrate already fused with a ubiquitin moiety on its N-terminus, which targets it for proteasomal degradation [10], and subsequent work identified *TRIP12* as the homologous protein to function in the ubiquitin fusion degradation pathway in mammalian cells [11]. Since then, *TRIP12* has been found to be involved in DNA damage response, oncogenic stress, cell cycle, and neurodegeneration [12,13,14,15]. Interestingly, *TRIP12* regulates response to PARP inhibitors in breast cancer cells [16] and we have shown that the genetic inhibition of *TRIP12* leads to the stabilization of *FBW7* protein and enhanced cell death in response to Taxol treatment [17]. However, the precise molecular details of enhanced cell death in response to Taxol in *TRIP12* deficient cells is not known. 

In this study, we show that *TRIP12* is required for exit from mitosis during mitotic block induced by Taxol. The *TRIP12*-deficient cells were arrested in mitosis and failed to re-enter cell cycle efficiently upon release from mitotic block. This was largely dependent on *FBW7* since in *FBW7*/*TRIP12* double knockout cells these effects were completely normalized to wildtype levels. Interestingly, the stable reconstitution of *FBW7* lysine-to-arginine double mutant, which is resistant to proteasomal degradation, in *FBW7* knockout cells leads to sharp reduction of *MCL-1* protein and culminates in enhanced cell death by Taxol. Finally, the *TRIP12*/*FBW7*/*MCL-1* axis is well preserved in ovarian cancer but not in breast cancer cells since siRNA-mediated depletion of *TRIP12* sensitized those cells to Taxol in an *FBW7*-dependent manner. Thus, our data provide strong evidence that *TRIP12* is essential for efficient mitotic arrest induced by Taxol and this effect is mediated via *FBW7*/*MCL-1* proteins.

## 2. Results

### 2.1. Enhanced Mitotic Degradation of MCL-1 in the Absence of TRIP12

We previously showed that genetic depletion of *TRIP12* stabilizes tumor suppressor protein *FBW7* and reduces CyclinE and *MCL-1* protein levels in HEK293 and HCT116 cells [17]. To check if the deletion of *TRIP12* affects *FBW7* protein and its substrates in other cell lines, we depleted *TRIP12* using two different short interfering (si)-RNA in U2OS cells. Consistent with our previous findings, *TRIP12* knockdown stabilizes *FBW7* protein and *FBW7* substrate CyclinE and *MCL-1* are downregulated (Figure 1). Additionally, the mammalian target of rapamycin (mTOR), another *FBW7* substrate, was also downregulated while c-Jun protein was increased in *TRIP12* knockdown U2OS cells (Figure 1). All other *FBW7* substrates including Notch1, c-MYC, HSF1, and SREBP1 were unaffected in *TRIP12* knockdown cells (Figure 1A and Appendix A). The preference of substrates by *FBW7* in *TRIP12*-depleted cells is not governed by *FBW7* dimerization because *FBW7* dimerization is largely unaffected in *TRIP12* knockdown cells (Appendix A). Because *MCL-1* is specifically targeted for proteasomal degradation by *FBW7* in mitosis [6], we checked the possibility that *TRIP12* regulates *MCL-1* levels in mitosis. Indeed, *MCL-1* degradation is enhanced in the absence of *TRIP12* in cells blocked in mitosis by releasing in nocodazole after a double thymidine block in two different cell lines (Figure 1B,C). We confirm the mitotic arrest by Western blots against the phospho-Histone 3 protein which is a bona fide marker of mitotic cells (Figure 1B,C). Next, we tested the possibility that *FBW7* protein stability is regulated by *TRIP12* in mitosis in HEK293 cells stably transfected with a control and *TRIP12* specific short hairpin (sh)-RNA (Appendix A). Consistent with previous findings [18], *FBW7* protein stability is not regulated by cell-cycle stages and *TRIP12* knockdown stabilized *FBW7* protein throughout the cell cycle. These results suggest that *FBW7* substrate specificity in *TRIP12*-depleted cells is context dependent and that *MCL-1* degradation is enhanced in the absence of *TRIP12* specifically in mitosis. 

### 2.2. TRIP12 Controls Exit from Mitosis via Negative Regulation of FBW7

Loss of function *FBW7* mutations leads to *MCL-1* protein accumulation which culminates in chemotherapy resistance of cancer cells [6]. We previously showed that targeting *TRIP12* enhances Taxol-induced cell death in colorectal cancer [17]. To understand the precise molecular mechanism of Taxol-induced death in those cells, we performed cell-cycle analyses of asynchronized as well as Taxol treated HCT116 cells. In comparison to wildtype cells, *TRIP12* knockout causes subtle increase in asynchronously growing mitotic cells (Figure 2A,B), indicative of enhanced proliferation or inability of the cells to exit mitosis efficiently. As expected, Taxol treatment led to the sharp induction of mitotic cells which was significantly increased in the absence of *TRIP12* (Figure 2A,C). Similar results were obtained in additional HEK293 cells where the mitotic arrest was more pronounced in *TRIP12* knockout cells compared to wildtype cells (Appendix A). The increase in mitotic cells in asynchronous as well as Taxol-arrested *TRIP12* knockout cells was near completely normalized to wildtype levels in *TRIP12*^−/−^*FBW7*^−/−^ (double knockout) cells. These results suggest two possibilities, either *TRIP12* is required for exit from mitosis during Taxol-induced mitotic arrest or *TRIP12* knockout cells enter mitosis faster than the controls as previously suggested [13]. 

To resolve this apparent bias, we synchronized HCT116 cells in mitosis and then released the cells in fresh media without mitotic blocker nocodazole. Strikingly, *TRIP12* deletion delayed exit from mitosis compared to wildtype cells, an effect which was completely normalized to wildtype levels in double knockout cells (Figure 3A,B). Consistent with that, a low dose of Taxol but not of other chemotherapy drugs cisplatin or etoposide, enhanced the cleaved-Caspase-7/total-Caspase-7 ratio in *TRIP12*-depleted cells compared to wildtype cells (Figure 3C), indicative of higher cell death. This effect was also normalized in double knockout cells to wildtype levels (Figure 3C). Finally, to test if *TRIP12* deletion sensitizes cells to Taxol-induced apoptosis, we generated stable cell lines expressing a lentivirus-mediated apoptosis reporter using a previously published Caspase activable GFP (CAGFP) expression plasmid [19]. When the cells are committed to apoptosis, the DEVD peptide is cleaved by caspases and allows for the expression of GFP and monitoring of early apoptosis via immunofluorescence or FACS (Figure 3D). As expected, Taxol treatment enhanced the number of GFP positive cells well above the background in wildtype cells which was roughly doubled in *TRIP12* knockout cells, an effect completely blunted by siRNA-mediated knockdown of *FBW7* (Figure 3E). Thus, our data suggest that *TRIP12* is required for efficient exit from mitosis and *TRIP12* knockout sensitizes HCT116 cells to apoptosis and these effects are largely mediated via *FBW7*. 

### 2.3. Mutating FBW7 Recognizable GSK3β Phosphodegron on MCL-1 Reverses Mitotic Arrest in TRIP12^−/−^ Cells

*FBW7* recognizes its substrate by their phosphorylated residues within a GSK3β phosphodegron. To check if negative regulation of *MCL-1* by *FBW7* is mediated through GSK3β-mediated phosphorylation, we mutated Ser159/162 and Thr163 in *MCL-1*’s phosphodegron (Figure 4A). The triple mutation not only stabilizes *MCL-1* (3A-mutant) protein in wildtype cells but also stabilizes the reduced *MCL-1* protein levels in *TRIP12* knockout cells similar to 3A mutant in wildtype cells as judged by Western blotting and normalization of *MCL-1* blots to GFP control blots (Figure 4B), thus confirming the requirement of GSK3β phosphorylation-mediated *FBW7* targeting of *MCL-1* protein degradation in wildtype as well as *TRIP12* knockout cells [6]. To test if enhanced degradation of *MCL-1* by *FBW7* in *TRIP12* knockout cells is responsible for mitotic arrest, we overexpressed *MCL-1* wildtype and 3A-mutant plasmids in *TRIP12* wildtype and knockout cells, synchronized those cells in mitosis, released them from mitotic arrest for 3–4 h, and then performed the cell-cycle analysis. As previously seen (Figure 3A,B), *TRIP12* knockout cells exited mitosis less efficiently than wildtype cells when wildtype *MCL-1* was overexpressed (Figure 4C). However, *MCL-1* 3A-mutant leads to more efficient exit from mitosis in both *TRIP12* wildtype and knockout cells. Thus, our data suggest that *TRIP12*/*MCL-1* axis is required for efficient mitotic exit, and *FBW7*-mediated proteasomal degradation of phosphorylated *MCL-1* may block this effect.

### 2.4. FBW7 Ubiquitylation Resistant Mutant Reduces MCL-1 Protein Levels and Sensitize HCT116 Cells to Taxol

*TRIP12* depletion stabilizes *FBW7* protein and sensitizes cancer cells in a *FBW7*-dependent manner (Figure 1A) [17]. In addition to *FBW7* regulation, *TRIP12* is involved in numerous biological functions [12,13,14,15]. To unequivocally establish that the enhanced *FBW7* function in HCT116 cells is responsible for Taxol-induced cell death, we sought to use an alternate approach. We previously showed that the *FBW7* K404/K412R double-point mutant is strongly stabilized due to its inability to autoubiquitylate itself on two crucial lysine residues essential for *FBW7* protein degradation [17]. First, we cloned *FBW7* wildtype and *FBW7* K404/412R (2R) mutant in a protein stability reporter plasmid and overexpress the two plasmids in HEK293FT cells. Consistent with previous findings, we noticed the sharp accumulation of the *FBW7* 2R-mutant compared to *FBW7* wildtype protein (Appendix A). Next, to check if *FBW7* stabilization alone would sensitize cancer cells and mimic *TRIP12* deletion, we used lentivirus-mediated overexpression of wildtype and K404/412R *FBW7* (2R) mutant in HCT116*^*FBW7*^*^−/−^ cells which are otherwise resistant to Taxol (Figure 5A) [6,17]. As expected, the *FBW7* 2R-mutant is strongly stabilized in HCT116*^*FBW7*^*^−/−^ cells and *MCL-1* protein levels were sharply reduced in those cells compared to HCT116*^*FBW7*^*^−/−^ cells overexpressing wildtype *FBW7* (Figure 5B). Finally, *FBW7* 2R-mutant and not *FBW7* wildtype overexpression sensitized HCT116 cells to increasing doses of Taxol (Figure 5C) and provides proof-of-concept evidence that suggests mitigating *FBW7* protein levels in chemotherapy resistance might be an interesting therapeutic target particularly in cancer patients with *FBW7* wildtype genotype.

### 2.5. Targeting TRIP12 Sensitizes Ovarian but Not Breast Cancer Cells to Taxol-Induced Cell Death

To this end, we described a pathway which could be therapeutically exploited for targeting cancers that are otherwise resistant to anti-mitotic chemotherapy. However, Taxol is not the standard of care in treatment of colorectal cancer in clinics. Thus, to test whether *TRIP12* inhibition will sensitize other cancer cells to Taxol, we knockdown *TRIP12* in ovarian cancer cells and treated those cells with increasing doses of Taxol. Strikingly, *TRIP12* depletion strongly sensitized *FBW7* wild type ovarian cancer cells whereas *FBW7*^R505L^ mutant cell line was resistant to Taxol treatment regardless of *TRIP12* status (Figure 6A–C). Consistent with previous results, *TRIP12* knockdown reduced *MCL-1* protein levels in *FBW7* wild type OVCAR3 and TOV112D cells but not in mutant SKOV3 cells (Figure 6D). Thus, our data suggest that *TRIP12* targeting might sensitize *FBW7* wild type ovarian cancer cells to Taxol therapy.

Next, we tested whether the depletion of *TRIP12* in breast cancer cells could achieve similar synergy with Taxol as witnessed with ovarian cancer cells. The triple negative breast cancer BT549 responded weakly to increasing doses of Taxol as judged by LDH cytotoxicity assay (Figure 6E). Although MCF7 cells were more sensitive to Taxol compared to BT549, *TRIP12* knockdown marginally enhanced cell death at very high doses of Taxol in those cells (Figure 6F). Interestingly, *TRIP12* knockdown in both BT549 and MCF7 cells did not affect *MCL-1* protein levels (Figure 6G), providing a possible explanation as to why *TRIP12* deletion could not sensitize those cells to low doses of Taxol. Finally, to test the hypothesis that high *MCL-1* levels in breast cancer cells help those cells escape mitotic cell death, we used siRNA-mediated targeting of *MCL-1* in Taxol resistant BT549 cells and study their response to Taxol (Appendix A). Indeed, siRNA-mediated *MCL-1* depletion strongly sensitized BT549 cells to Taxol (Appendix A), these results are consistent with previous findings [20] and highlight the importance of considering *MCL-1* protein levels for treatment of triple breast cancer.

## 3. Discussion

Cancer chemotherapy resistance is a major hurdle in patients’ treatment. However, despite producing nominal benefits in most cases, chemotherapy has been widely used for cancer treatment for decades. The philosophy behind this relentless practice is ‘one treatment for all’ cancer patients. Such an approach often ignores the tumor heterogeneity, patients’ genetic background, de novo resistance to available treatments, cancer stages, molecular diversity, and aggressive metastatic disease. On the contrary, an alternate approach by stratifying patients into groups based on their genetic or molecular profiles might provide a more effective way of cancer management [21]. Thus, understanding the molecular mechanisms of chemotherapy resistance can provide relevant clinical or molecular biomarkers that may help predict chemotherapy response.

Loss of *FBW7* function mutation is long associated with cancer chemotherapy resistance and aggressive phenotypes [22]. *FBW7* facilitates phosphorylation-dependent proteasomal degradation of multiple proto-oncogenic molecules [5]. In the absence of *FBW7* activity, several *FBW7* substrates accumulate including antiapoptotic *MCL-1*. High *MCL-1* levels not only prevent cell death of *FBW7* mutated cells due to unusually high oncogenic signaling from c-MYC, CyclinE, and c-JUN proteins, but also provide a mechanism to escape chemotherapy-induced cell death [6]. Interestingly, the majority of *FBW7* mutations in human cancers are heterozygous with at least one wildtype allele retained by the patient [5]. Additionally, others and we have shown that *FBW7* protein is downregulated independent of its mutational status in many cancer patients [8]. Thus, it is reasonable to believe that chemotherapy resistance can be blocked by enhanced *FBW7* activity by directly interfering with its protein turnover. However, such a hypothesis has not been tested until recently [17]. 

We previously established E3 ubiquitin ligase *TRIP12* to be a negative regulator of *FBW7* protein [17]. Inhibition of *TRIP12* not only stabilizes *FBW7* protein but also leads to downregulation of *MCL-1* protein (Figure 1). The enhanced *MCL-1* degradation in *TRIP12* knockout cells is largely carried out during mitosis (Figure 1) which culminates in aberrant exit from mitosis and affects cell-cycle re-entry, particularly in response to Taxol, a mechanism most likely responsible for enhanced cell death in *TRIP12* knockout cells by Taxol. These results have broader clinical implications, since *FBW7* protein is downregulated in cancer and *TRIP12* may provide a target for pharmacological intervention to restore *FBW7* activity. Additionally, *TRIP12*/*MCL-1* expression might be a useful biomarker for texane-based chemotherapy response in some but not all cancers. 

Although we find consistent downregulation of *MCL-1* protein in the absence of *TRIP12* in multiplate cell lines, exactly what defines substrate specificity of *FBW7* in *TRIP12* depleted cells is not clear. For example, previous work demonstrated that *FBW7* dimerization provides selectivity towards its substrates in a context dependent manner [5]. Yet, we do not find *FBW7* dimerization to be affected by *TRIP12* inhibition. Moreover, the majority of *FBW7* substrates were unaffected upon *TRIP12* inhibition including c-MYC, c-Jun, and Notch-1 in HEK293 cells [17]. Contrary to that, we find c-Jun accumulation in U2OS cells upon *TRIP12* inhibition (Figure 1A). These differences in c-Jun protein could be due to the tissue specific nature of *FBW7* activity towards its substrates or, alternatively, a result of enhanced gene expression and totally unrelated to protein stability.

Our data show remarkable downregulation of *MCL-1* protein and near complete reversal of Taxol resistance by overexpression of highly stabilized *FBW7* 2R-mutant which demonstrates the power of mitigating *FBW7* and *MCL-1* protein levels in human cancers for efficient chemotherapy response. This could be achieved by pharmacological targeting of *TRIP12* to increase endogenous *FBW7* protein level; however, to date, there are no specific inhibitors available for *TRIP12*. Chemical biology screens can be designed to scan and identify *TRIP12* specific inhibitors. One caveat to this approach is that *TRIP12* contains a highly conserved HECT domain, thus making it largely difficult to specifically inhibit its activity without affecting the function of broader HECT family members involved in diverse cellular functions. Alternatively, small molecule inhibitors of *MCL-1* can be exploited against cancer and chemotherapy resistance [23]. Interestingly, some of those inhibitors are already tested against acute myeloid leukemia and Hodgkin’s lymphoma in clinical trials [23]. Once completed, these studies may shed some invaluable light on the utility of *MCL-1* inhibitors for cancer treatment and might encourage clinical trials against solid cancers.

Finally, our data show that colorectal and ovarian cancer cells can be sensitized to Taxol by genetic inhibition of *TRIP12* and this effect is largely dependent on *FBW7* and *MCL-1* proteins. However, this genetic interaction was not seen in breast cancer cells because *TRIP12* inhibition in those cells did not enhance Taxol-mediated cell death. Interestingly, *MCL-1* amplifications are more frequent in breast cancer compared to colorectal or ovarian cancer which will ultimately nullify the impact of *TRIP12* inhibition in those cells. Of note, we used two different breast cancer cell lines MCF7A (ER+/PR+) and BT549 (triple negative), and MCF7A cells that expressed much less *MCL-1* protein compared to BT549, were more sensitive to Taxol (Figure 6F). Deletion of *TRIP12* in MCF7A cells had negligible effect on Taxol-mediated cell death; albeit only at higher doses did it marginally enhance cell death. However, BT549 that were largely resistant to *TRIP12* inhibition and Taxol treatment, were readily sensitized to Taxol by siRNA-mediated *MCL-1* knockdown. Thus, our data highlight the importance of stratifying breast cancer patients on their *MCL-1* expression for Taxol therapy as previously suggested [20]. Importantly, targeting *MCL-1* may provide an alternate treatment option for triple negative breast cancer patients in combination with Taxol. Importantly, our study provides evidence that *TRIP12*/*FBW7*/*MCL-1* axis is a major determinant of Taxol-mediated mitotic arrest and *MCL-1* protein is at the heart of this genetic interaction (Figure 7).

## 4. Methods and Materials

### 4.1. Cell Lines

All cell lines were obtained from the cell services of the Francis Crick Institute (London, UK) and were maintained as per the guidelines from ATCC or previously reported [17]. 

### 4.2. Site Directed Mutagenesis

*MCL-1* serine to alanine mutants were made using the Quick-change lightning site-directed mutagenesis kit (Agilent, Stockport, UK). Mutated plasmid clones were validated by Sanger sequencing, and then used for subsequent overexpression in mammalian cells followed by Western blot or cell-cycle analysis.

### 4.3. Western Blot Assays

Immunoblotting was carried out as previously described [24]. Briefly, cells were harvested, washed, lysed in 1x cell lysis buffer (#9803, CST), and ran on 7.5 or 10% Tris-HCL SDS-PAGE gels. After the wet-transfer of proteins on nitrocellulose membrane, the membranes were blocked for 1 h at room temperature in 5% non-fat dry milk, and then incubated in primary antibodies overnight at 4 °C.

### 4.4. Antibodies

Antibodies used for Western blotting were anti-*FBW7*α (#A301-720A), and anti-*TRIP12* (#301-814A) from Bethyl laboratories (US), anti-vinculin (#V9131) from Sigma-Aldrich (St. Louis, MA, USA), anti-Actin HRP conjugated (#ab-49900), anti-GAPDH (#ab9485), anti-c-MYC-Y69 (#ab-32072) from Abcam (Cambridge, UK), anti-CyclinE (#sc-481) from Santa Cruz Biotechnology (Dallas, TX, USA), anti-*MCL-1* (#54539) and anti-Cleaved caspase-7 (#9491) from Cell signaling technology (Denver, CO, USA).

### 4.5. Cell Viability Assays

Cell viability assays were performed using Promega’s Cell-titer blue cell viability assay as per the vendor’s instructions. Briefly, 5000 cells/well were plated on a 96-well plate in triplicates and allowed to adhere overnight. In the next morning, indicated doses of Taxol were added to the cells and plates were further incubated up to 72 h. After 72 h, CTB reagent was diluted in complete DMEM and added on the cells; then the plate was further incubated between 30 to 60 min and read on a TECAN plate reader as per the kit’s (Promega, G8081) protocol.

### 4.6. LDH Cell Cytotoxicity Assays

Cell Cytotoxicity was measured by LDH kit (#11644793001, Roche/Genentech, Basel, Switzerland). Briefly, cells were treated with Taxol as above and 72 h after Taxol treatment, 100 μL of cell-free supernatant was transferred into another 96-well microplate, spun to remove cell debris, and incubated with 100 μL of LDH-assay reaction mixture for 15–30 min at 37 °C in humidified incubator set at 5% CO_2_. LDH released in the media due to cytotoxicity was able to reduce NAD+ to NAD+H+ which in turn converted the yellow tetrazolium salt to formazan red salt. Absorbance of the color generated was measured by TECAN microplate reader at 492 nm and 620 nm.

### 4.7. Cell Cycle Analysis

Cell-cycle analysis was performed as reported before [25]. For synchronizing cells in mitosis, cells were blocked in 2 mM thymidine for 24 h. After 24 h, cells were released in complete DMEM 10% FBS media for 3 h and then arrested in mitosis by adding 100 ng/mL nocodazole overnight. After overnight mitotic block, cells were washed and collected in ice chilled PBS, fixed in 70% ethanol, washed 2X in PBS, treated with 20 μg/mL RNase H, and stained with 50 µg/mL propidium iodide overnight. Cell-cycle profiles were obtained on BD CSampler^TM^ plus. For Western blots of *MCL-1* in mitosis, cells were synchronized by double thymidine block and released in 100 ng/mL nocodazole for the indicated times.

### 4.8. Generation CA-GFP Apoptosis Reported Plasmid and Cell Lines

CA-GFP plasmid was previously published [19]. We used CA-GFP plasmid sequence from addgene plasmid (#32748) to design a bicistronic expression vector expressing CA-GFP under the CMV promotor separated by an internal ribosome entry site (IRES)-mediated expression of dTomato reporter protein. The plasmid was sourced from vectorbuilder.com. Lentivirus preparation, transduction, and cell selection was performed as before [26]. 

## Figures and Tables

**Figure 1 cancers-15-00505-f001:**
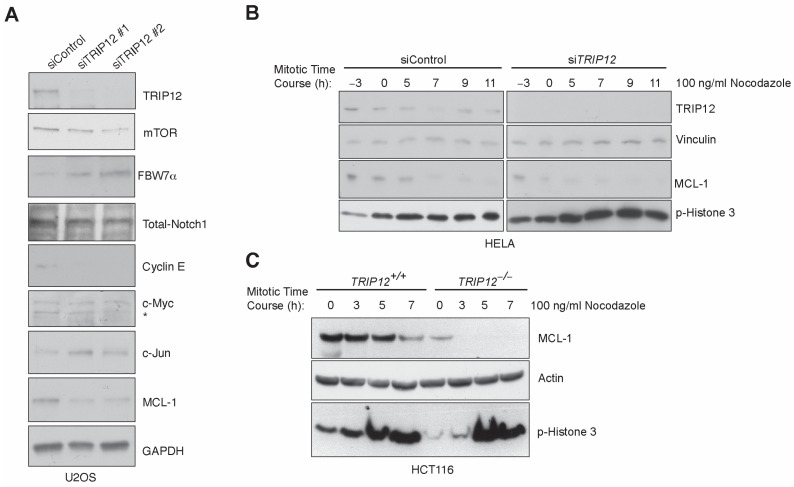
*FBW7* protein is stabilized and its substrates mTOR, CyclinE, and *MCL-1* are downregu–lated in *TRIP12* depleted cells. (**A**), Western blot validation of *FBW7* protein stability and substrates on U2OS cell lysates transfected with two independent siRNAs targeting *TRIP12* gene compared to a non-targeting control. Note: * denotes non-specific immunoreactive band. (**B**,**C**) Immunoblot showing decreased *MCL-1* levels in HELA and HCT116 cells transfected with *TRIP12* siRNA followed by a double thymidine block and subsequent release in 100 ng/mL nocodazole for indicated times relative to ‘0’ h, which is an approximate mitotic onset time after release from thymidine block.

**Figure 2 cancers-15-00505-f002:**
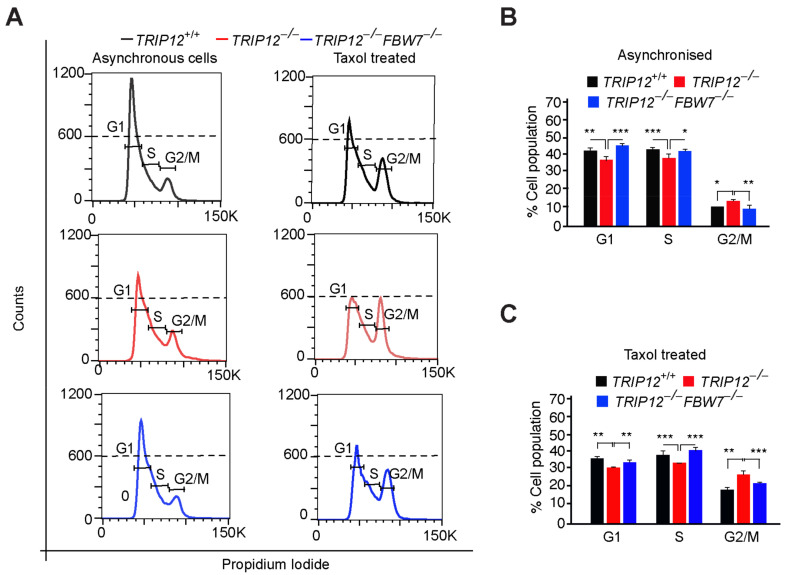
Enhanced mitotic arrest in *TRIP12*-depleted cells: (**A**) Histogram of asynchronously grow–ing and Taxol treated HCT116 cells with indicated genotypes. Following treatment with Taxol, cells were collected, fixed, and stained with Propidium iodide (PI), and the DNA content was analyzed by FACS. (**B**,**C**) Quantification of fluorescence intensity of the PI-stained DNA in a population of cells within distinct cell-cycle phases from A. Mean ± S.D. of 3 independent experiments is shown. * *p* < 0.05, ** *p* < 0.01, and *** *p* < 0.001.

**Figure 3 cancers-15-00505-f003:**
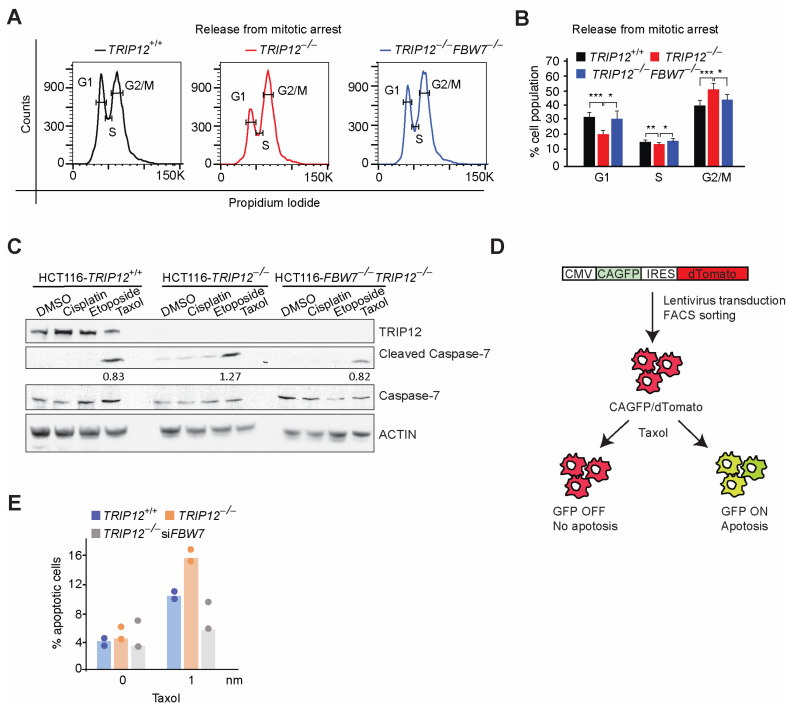
Aberrant release from mitosis in *TRIP12*-depleted cells: (**A**), Histogram of cells with indi–cated genotypes, synchronized in mitosis and released in normal media. (**B**) Quantification from experiments in (**A**). (**C**) Western blot for indicated antibodies showing cleaved caspase-7/total caspase-7 ratio in cells from indicated genotypes. (**D**) Schematic for generation of a CA-GFP fluorescent apoptosis reporter. (**E**) Quantification of FACS data from stable cell lines expressing lentivirus-mediated CA-GFP in cells from indicated genotypes. * *p* < 0.05, ** *p* < 0.01, and *** *p* < 0.001.

**Figure 4 cancers-15-00505-f004:**
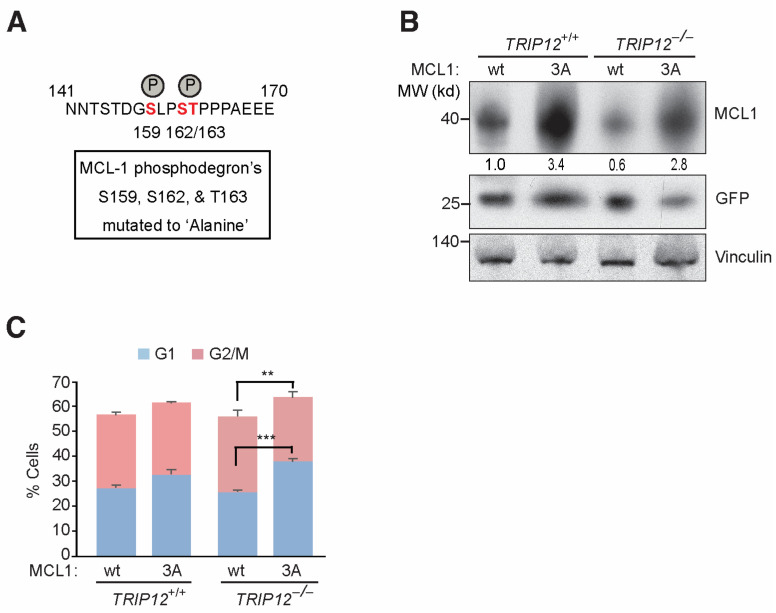
Efficient release from mitotic arrest in cells overexpression *MCL-1* 3A-mutant. (**A**) Sche–matic of *MCL-1*’s GSK3β phosphodegron mutated to alanine. (**B**) Western blot for indicated antibodies in cells overexpressing *MCL-1* wildtype and 3A-mutant. Numbers under *MCL-1* blot denotes densitometric quantification normalized to transfection control GFP. Vinculin antibody is used as a loading control. (**C**) *MCL-1* wild type and mutant overexpressing mitotically synchronized *TRIP12* depleted cells subjected to cell-cycle analysis after 3 HR of release from mitotic arrest assessed by flow cytometry. ** *p* < 0.01, and *** *p* < 0.001.

**Figure 5 cancers-15-00505-f005:**
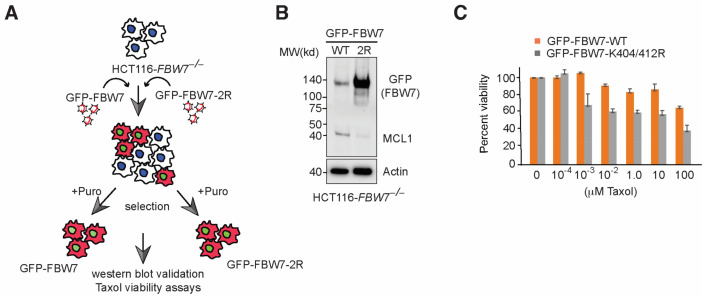
*FBW7* K404/412R mutant reduces *MCL-1* protein and sensitize HCT116-*FBW7*^−/−^ cells to Taxol. (**A**) Schematic of generation of *FBW7* wildtype and K404/412R mutant cell lines. (**B**) Immunoblot showing *FBW7* K404/412R stability and downregulation of *MCL-1* in HCT116-*FBW7*^−/−^ cells in comparison to wild type controls. (**C**) Cell titer blue viability assay quantification from HCT116-*FBW7*^−/−^ overexpressing *FBW7* wild type and 2R mutant cells treated with increasing doses of Taxol. Mean of 3 independent experiments is shown as percent viability.

**Figure 6 cancers-15-00505-f006:**
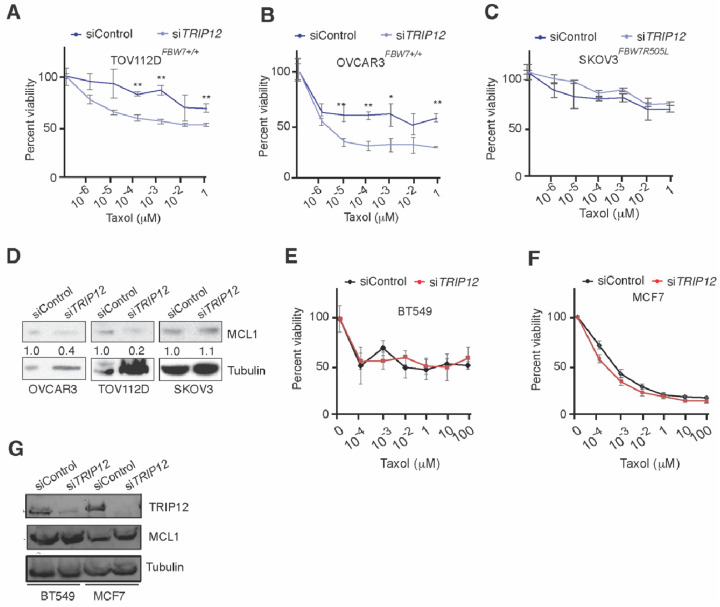
Analyzing the role of *TRIP12* inhibition in sensitizing ovarian and breast cancer cells to Taxol. (**A**–**C**) Cell titer blue viability assay quantification in ovarian cancer cells of indicated geno–types treatment with indicated doses of Taxol. (**D**) Western blot for indicated antibodies in ovarian cancer cells incubated with siControl or si*TRIP12*. Numbers under the MCL1 blot represents densitometric quantification normalized to loading control α-Tubulin. (**E**,**F**) Cell titer blue viability assay quantification in breast cancer cells treated with indicated doses of Taxol. (**G**) Western blot for indicated antibodies in ovarian cancer cells incubated with siControl or si*TRIP12*. Note: * = *p* < 0.05 and ** = *p* < 0.01.

**Figure 7 cancers-15-00505-f007:**
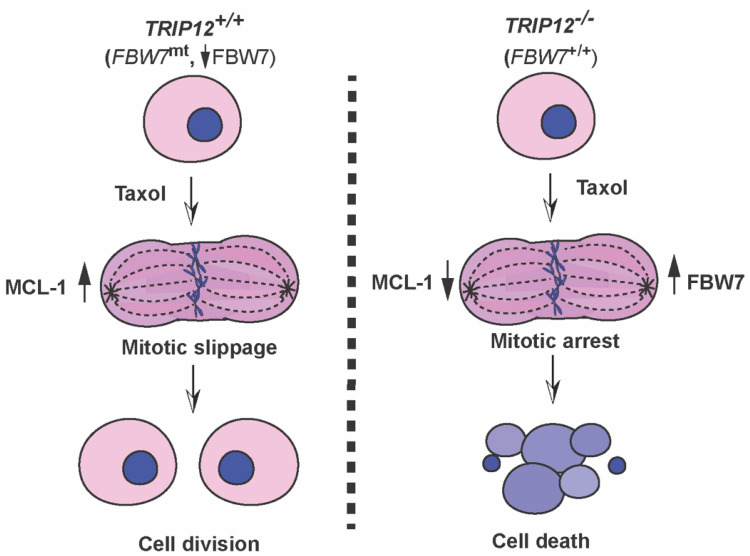
*TRIP12* inhibits mitotic arrest in response to Taxol.

## Data Availability

Data sharing not applicable.

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
