# Peer review of "E3 Ubiquitin Ligase TRIP12 Controls Exit from Mitosis via Positive Regulation of MCL-1 in Response to Taxol"

_cancers, 2023, doi:10.3390/cancers15020505_

Round 1

Reviewer 1 Report

In the manuscript authors have elucidated the mechanistic details of Taxol resistance in the cancer cells. Previously it was known that FBW7 degradation which promote Taxol resistance in the cancer cells is promoted by TRIP12. It was also known that FBW7 promotes degradation of apoptosis inhibitor MCL1 which sensitizes cells for Taxol mediated cell cycle arrest and apoptosis. It was also known that the taxol resistance in cancer cells happens through inactivation of FBW7. However, previously it was not shown that TRIP12 dependent Taxol resistance occur through TRIP12-FBW7-MCL1 axis. Authors have shown that the TRIP12 knock out cells are arrested at M phase whereas TRIP12/FBW7 double knock outs are rescued from such arrest. Finally, authors have shown that while the knock down of TRP12 in ovarian cancer cells are making those cells responsive to Taxol, such effect is not seen in beast cancer cells, and this could be due to non-degradation of MCL1 proteins in breast cancer cells in spite of TRP12 knock down.
The manuscript has made advancement in our understanding of Taxol resistance in cancer cells and its mechanistic variations in different cancer lineages.   However, the manuscript needs major improvements before it can be accepted for publication. The reviewer has following suggestions for the improvement of the manuscript.
General comments:
The manuscript must be checked thoroughly for proper figure numbers and other formatting errors and language overall.
Specific comments:
1.    Fig. 1A: Previous study by the authors have shown no effect on c-Myc upon TRIP 12 suppression. It is not clear here what is the effect of TRIP 12 knock down on c-Myc expression. Why is c-Jun increased compared to control? These observations must be discussed adequately.
2.    Fig. 1B: What is denoted by ‘M’ in the figure? It must be mentioned in the figure legend.
3.    Fig 1C: Observation on p-Histone 3 and its inference is not discussed in the text. It must be discussed.
4.    Line # 110 and 112. Figure numbers are wrong.
5.    Line # 123. While from the figure 3C it’s appearing that it is the ratio of cleaved Caspase/ total caspase is measured, the text describes otherwise. This must be clarified.
6.    Are the data shown in the Fig. 3E statistically significant?
7.    Figure 4B: The level of 3A mutant in TRP12 knock out background is clearly lower compared to the WT cells. How authors explain such difference when 3A mutant suppose to be resistant to degradation. In this regard it is not clear what is meant by the phrase “back to wild type protein levels” in line # 150 in the text. Further, how was the exogenous MCL1 differentiated from the endogenous MCL1?
8.    Figure 5B: How was it confirmed that the higher level of FBW7-2R mutant was due to reduced degradation and not due to higher expression of the protein? More confirmatory experiment would be to analyse the degradation kinetics of the FBW7-2R mutant. Further, how was the exogenous FBW7 differentiated from the endogenous FBW7?
9.    Fig. 6D: Quantification of the Western data will make the observation more convincing.
10.    Fig. 6E & 6F: While these figures and Fig. 6 A, B, and C all describe cell survival/cytotoxicity data, it is not clear why authors chose to represent these data in different manner in breast cancer cells compared to the ovarian cancer cells.
11.    Line # 209: Breast cancer cells are mentioned as ovarian cancer cells.

Author Response

Reviewer 1:

Overall Feedback: In the manuscript authors have elucidated the mechanistic details of Taxol resistance in the cancer cells……The manuscript has made advancement in our understanding of Taxol resistance in cancer cells and its mechanistic variations in different cancer lineages.  

Answer: We are thankful to the reviewer for their clarity of understanding the background of our work and their overall positive feedback on our manuscript. In agreement with the reviewer’s feedback of manuscript requiring a thorough revision we have addressed the comments raised by the reviewer in a point-by-point manner.

Comment #1: Fig. 1A: Previous study by the authors have shown no effect on c-Myc upon TRIP 12 suppression. It is not clear here what is the effect of TRIP 12 knock down on c-Myc expression. Why is c-Jun increased compared to control? These observations must be discussed adequately..

Answer: We thank the reviewer for pointing out the discrepancy between the c-Jun findings in current manuscript compared with the previously published work. Overall, our conclusion for c-Myc protein is that it’s not affected by the TRIP12 deletion, however c-Jun is showing robust accumulation upon TRIP12 deletion in U2OS cells. We believe that this is largely due to tissue specific nature of FBW7 regulation of its substrates. For example, c-Jun protein stability is regulated by multiple E3 ubiquitin ligases across different tissues. In previous work, we used HEK293 cells which are of embryonic origin while in this work we used U2OS cells which is an osteosarcoma cell line. We have discussed this in discussion on page 6 (paraph 2) for better clarity.

Comment #2: Fig. 1B: What is denoted by ‘M’ in the figure? It must be mentioned in the figure legend?

Answer: We thank the reviewer for pointing this out. ‘M’ denotes approximate time of mitotic onset. We have changed this to ‘0’ in the figure now and added explanation for better clarity in Figure legend 1 on page 11.

Comment #3: Fig 1C: Observation on p-Histone 3 and its inference is not discussed in the text. It must be discussed.

Answer: We thank the reviewer for pointing this out. We have now added the text for this observation in main text in results on page 3 and paragraph 1.

Comment #4: Line # 110 and 112. Figure numbers are wrong.

Answer: We have gone through with this twice and do not find any discrepancy between the results text and figure numbers. However, we have gone through the entire manuscript carefully and corrected some mislabelling of the figures in other places.

Comment #5: Line # 123. While from the figure 3C it’s appearing that it is the ratio of cleaved Caspase/ total caspase is measured, the text describes otherwise. This must be clarified..

Answer: We are thankful to the reviewer for pointing out this mistake. Indeed, the ratio shown is of cleaved Caspase/ total caspase and this has been corrected in main texts results on page 3 paragraph 2.

Comment #6: Are the data shown in the Fig. 3E statistically significant? 

Answer: The data shown in figure 3E is the mean of 2 different experiments. It is not possible to perform statistics on this data. We have now replaced the bar graph with dot plot for better clarity.

Comment #7: Figure 4B: The level of 3A mutant in TRP12 knock out background is clearly lower compared to the WT cells. How authors explain such difference when 3A mutant suppose to be resistant to degradation. In this regard it is not clear what is meant by the phrase “back to wild type protein levels” in line # 150 in the text. Further, how was the exogenous MCL1 differentiated from the endogenous MCL1?

Answer: We thank the reviewer for raising this important point. In Fig 4B, the MCL-1 protein is overexpressed exogenously and to control transfection artefacts we use co expression of a GFP expressing plasmid. Thus, GFP expression is a direct readout of transfection efficiency. When normalised to GFP, MCL-1 3A mutant in TRIP12 knockout cells is restored to MCL-1 3A mutant levels in TRIP12 WT cells. We have added the densitometric ratio under the blots in Fig 4B and rephrased the text on page 4 paragraph 2 for better clarity and added this information in Figure legend of Fig 4B on page

Comment #8: Figure 5B: How was it confirmed that the higher level of FBW7-2R mutant was due to reduced degradation and not due to higher expression of the protein? More confirmatory experiment would be to analyse the degradation kinetics of the FBW7-2R mutant. Further, how was the exogenous FBW7 differentiated from the endogenous FBW7?

Answer: We thank the reviewer for raising this technical point. We previously showed that FBW7 2R (K404/412R) double point mutant is resistant to proteasomal degradation (Khan et al 2021). In Fig5B we cloned the FBW7 2R mutant to a protein stability reporter plasmid plpGPS_dsRED_IRES_eGFP (Khan et al 2021). In this plasmid, a single mRNA overexpresses an internal control dsRED and a GFP fused protein. Thus, GFP-FBW7 protein expression can be normalized to dsRED to compare the stability of FBW7-2R versus FBW7 wildtype. As suggested by the reviewer, we address this issue by overexpressing the two plasmids in HEK293FT cells. Consistent with previous results, FBW7 2R mutant shows stronger protein stability compared to the FBW7 wildtype. These results are now added as Supplementary Figure 2B and the text is modified on page 4 paragraph 3.

Comment #9: Fig. 6D: Quantification of the Western data will make the observation more convincing. 

Answer: We thank the reviewer for this suggestion and have now added the densitometric quantification under the blots in figure 6D and added this information in Figure legend of Figure 6.

Comment #10: Fig. 6E & 6F: While these figures and Fig. 6 A, B, and C all describe cell survival/cytotoxicity data, it is not clear why authors chose to represent these data in different manner in breast cancer cells compared to the ovarian cancer cells.

Answer: We thank the reviewer for this suggestion we have now corrected this and for the sake of consistency represented the data as ‘percent viability’ throughout the figure.

Comment #10: Line # 209: Breast cancer cells are mentioned as ovarian cancer cells.

Answer: We thank the reviewer for pointing out this error. We have now corrected this mistake in main text.

Reviewer 2 Report

In their study, Keyan et al. propose that the TRIP12 deletion leads to enhanced mitotic arrest and cell death in FBW7 and MCL-1 dependent manner in multiple cell lines, including colorectal and ovarian cancer but not in breast cancer. Altogether the authors provide interesting findings, with possible TRIP12/FBW7/MCL-1 axis, which may provide a therapeutic target to overcome Taxol-associated chemotherapy resistance in cancer. Overall, the manuscript would benefit substantially from a revision. At present, there is no data in the manuscript which can associate the findings with a putative pathophysiological relevance.

Major Points:

1-     Authors should use more than one cell line for each cancer type to show their significant findings.

2-     CRISPR Knockout for TRP12 with at least two gRNA is recommended.

3-     Authors should analyze whether the TRIP12/FBW7/MCL-1 axis affects cancer patients' overall or disease-free survival.  

Minor:

Figure 2 is misplaced in the manuscript.

Author Response

Reviewer 2:

Comment #1: In their study, Keyan et al. propose that the TRIP12 deletion leads to enhanced mitotic arrest and cell death in FBW7 and MCL-1 dependent manner in multiple cell lines, including colorectal and ovarian cancer but not in breast cancer. Altogether the authors provide interesting findings, with possible TRIP12/FBW7/MCL-1 axis, which may provide a therapeutic target to overcome Taxol-associated chemotherapy resistance in cancer. Overall, the manuscript would benefit substantially from a revision. At present, there is no data in the manuscript which can associate the findings with a putative pathophysiological relevance.

Answer:  We are thankful to the reviewer for critical analysis of our manuscript and their constructive input. In agreement with the reviewer’s feedback of manuscript requiring a thorough revision we have addressed the comments raised by the reviewer in a point-by-point manner.   

Comment #2: 1-     Authors should use more than one cell line for each cancer type to show their significant findings.

Answer:  We are thankful to the reviewer for making this point. Indeed, we have used several cell lines throughout the manuscript to support our findings. For example, MCL-1 western blot data in Figure 1 is reproduced in 3 different cell lines (U2OS, HELA, & HCT116). The taxol sensitivity data in Figure 6 is tested in 3 independent ovarian (SKOV3, OVCAR3, & TOV112D) and 2 independent breast cancer cell lines (MCF7A  & BT549) respectively. However, we chose HCT116 cells for main mechanistic studies for several reasons. First, we had previously shown that multiple colorectal cancer cells are sensitized to taxol treatment up on TRIP12 inhibition including HCT116 cells (Khan et al 2021). Second, the TRIP12/FBW7/MCL-1 axis is very well conserved in HCT116 cells. Finally, the availability of HCT116 TRIP12-FBW7 double knockout mutant cells which were generated previously means we could readily test our hypothesis directly in those cells. Nevertheless, we confirmed the mitotic arrest observation in an unrelated TRIP12 knockout cells and added this data to Supplementary Figure 2A. Our conclusion regarding the enhanced mitotic arrest in the absence of TRIP12 is further reinforced by this experiment.

Comment #3: CRISPR Knockout for TRP12 with at least two gRNA is recommended.? 

Answer:  We thank the reviewer for making this important observation. We agree with the reviewer that one should use more than a single gRNA for genetic inhibition of target genes for testing their hypotheses to control for off-target effects. However, more than often these genetic knockouts are created by conventional lentivirus mediated stable and constitutive overexpression of Cas9 and sgRNA, thus increasing the chances of off-target effects. In the case of HCT116 TRIP12 knockout cells used in this study, we generated these cells under the control of doxycycline inducible Cas9 combined with a chemically synthesized crRNA. Thus, allowing for transient overexpression of both Cas9 and crRNA reducing the chances of off-target effects due to constitutive induction of Cas9 and sgRNA. In the light of our previous and current work including use of multiple siRNA, shRNA. and CRISPR-mediated genetic inhibition of TRIP12 we believe that generating new CRISPR knockout cells is currently out of the scope of this work, particularly the time window of 20-days given to resubmit the revised manuscript is not compatible with such experiment. However, should the reviewer wish to see multiple CRISPR knockout cells, we are more than happy to proceed with this experiment in next round of revision.

Comment #4: Authors should analyze whether the TRIP12/FBW7/MCL-1 axis affects cancer patients' overall or disease-free survival.  

Answer:  We thank the reviewer for this critical point. To understand the role of TRIP12 in pathophysiology of cancer, we first detected the levels of TRIP12 gene expression in TCGA datasets from GEPIA.  However, we do not find any significant upregulation of TRIP12 gene expression in majority of cancer datasets available except DLBC, PAAD, and Thymoma (Figure). This suggests that TRIP12 is not a driver oncogene but might only be involved during the chemotherapy treatment phase of the disease. Consistent with that, to check if TRIP12/FBW7/MCL1 axis might have any impact on cancer patients’ survival, we first analysed the overall – and disease-free survival of ovarian and colorectal cancer patients using the TCGA datasets available on GEPIA2 platform. We do not find any significant survival differences between high and low TRIP12 expressing patients’ cohorts (data not shown). Because patients’ survival is not affected by TRIP12 expression, we believe that analyses of MCL1 and FBW7 in relation to patients’ survival is meaningless for this study because several other studies have already shown the effect of MCL1 and FBW7 gene expression on patients’ datasets.

Figure: Dot plot showing TRIP12 gene expression across multiple human cancer biopsies and surrounding normal tissue.

Finally, there could be multiple reasons why TRIP12 expression is not correlated with the patients’ survival. First, many of these patients might have MCL-1 amplifications which will render TRIP12 expression meaningless for this cohort. Second, underlying FBW7 mutations might be sufficient for aggressive disease and chemotherapy resistance since those mutations will directly lead to accumulation of MCL-1 protein. Third, the possibility of more than one chemotherapy resistance mechanisms in the cancer cohorts tested – as is often the case in many cancers. Last, TRIP12 may have cancer stage specific context dependent effect thus not having a pronounced effect on cancer patients’ survival.

Minor comments

Comment: Figure 2 is misplaced in the manuscript.

Answer: We thank the reviewer for pointing out this mistake. We have now corrected this on page 3 and referred Figure 2 in the main text.

Round 2

Reviewer 2 Report

The manuscript by Keyan et al. has improved substantially after the revision. The authors have provided a convincing explanation of most of the questions raised. I have a minor suggestion:

1-     As the inhibitor for TRIP12 is not available yet, the authors can use known MCL-1 inhibitors along with Taxol to show the synergy and their claim for targeting the TRIP12/FBW7/MCL-1 axis as therapeutics. The suggested experiment could further improve the quality of the manuscript.